# New Insights on Alternative Hosts of *Xanthomonas vasicola* pv. *vasculorum*, the Causal Agent of Bacterial Leaf Streak of Maize

Talita Vigo Longhi [1,2,*], Renata Rodrigues Robaina [1], Deived Uilian de Carvalho [1,2], Admilton Gonçalves de Oliveira [2], Rui Pereira Leite Junior [1] and Maria Isabel Balbi-Peña [2]

1 Instituto de Desenvolvimento Rural do Paraná-IAPAR/Emater (IDR-Paraná), Km 375 Celso Garcia Cid Road, Londrina 86047-902, Brazil
2 Departamento de Agronomia, Universidade Estadual de Londrina (UEL), Km 380 Celso Garcia Cid Road, Londrina 86057-970, Brazil
* Correspondence: talita.longhi@uel.br

**Abstract:** Bacterial leaf streak (BLS) of maize (*Zea mays*), caused by the bacterium *Xanthomonas vasicola* pv. *vasculorum* (*Xvv*), was first reported in Brazil in 2018. In this study, we evaluated 52 species of cultivated plants, cover crops, forage, and grasses that are used in succession or crop rotation with maize, and weeds with natural occurrence in maize-producing regions, to determine their potentials as alternative hosts for *Xvv*. We investigated (i) the pathogenicity of *Xvv* based on symptom development, (ii) epiphytic colonization of the bacterium in asymptomatic plants, and (iii) bacterial colonization in plant tissues using scanning electron microscopy (SEM) in symptomatic and asymptomatic species. Ten species, all belonging to the Poaceae family, presented symptoms after *Xvv* infection, including *Avena sativa* (cvs. IPR Afrodite and IPR Esmeralda), *A. strigosa* (cv. IPR 161), *Hordeum vulgare* (cv. BRS Cauê), *Oryza sativa* (cv. IPR 117), *Brachiaria brizantha* (Brizantha and cv. Marandu), *Digitaria horizontalis*, *D. insularis*, *Echinochloa colonum*, *Eleusine indica*, and *Sorghum arundinaceum*. Furthermore, epiphytic colonization by *Xvv* was observed in 23 asymptomatic species. Scanning micrographs revealed that *Xvv* cells and their aggregates were distributed throughout the leaf surface. In addition, bacterial colonization in the intercellular tissues of the substomatal chambers of white oat, black oat, and maize was observed across the tissue fractures. Despite showing typical symptoms of *Xvv* infection, SEM examination revealed evidence of *Xvv* colonization only on the leaf surface of rice. In asymptomatic species, such as rye, sorghum, and millet, a low number of bacterial cells were found on the leaf surface. However, no evidence of internal tissue colonization was observed in millet fractures, suggesting that *Xvv* survives only epiphytically in this species.

**Keywords:** *Zea mays*; bacterial disease; epiphytic colonization; scanning electron microscopy (SEM); pathogenicity; symptomatology





## 1. Introduction

Bacterial leaf streak (BLS) of maize (*Zea mays*), caused by the bacterium *Xanthomonas vasicola* pv. *vasculorum* (*Xvv*), was first reported in Brazil in 2018 [1]. However, this bacterial disease has previously been reported in maize-producing areas in the U.S. and Argentina [2–7]. In Brazil, BLS is present in several municipalities of the northern and western regions of the state of Paraná and has challenged agricultural researchers and Brazilian growers [7].

BLS symptoms in maize are initially characterized by the presence of small translucent, water-soaked spots and anasarca on leaves that expand along the interveinal tissues, forming elongated streaks with irregular and wavy edges [2,7]. The lesions range from an intense yellow to brown color, and afterwards become necrotic. Other plant species have also been reported as hosts for *Xvv*. In 1935, the occurrence of sugarcane (*Saccharum officinarum*) systemically infected with *Xvv* was reported in New South Wales, Australia [8]. This bacterium has also been reported in South Africa since the 1940s, occurring naturally in maize [9]. Currently, *Xvv* is present in several important crop-producing areas

worldwide [10–12]. Several plant species have been reported as hosts for *Xvv*, including *Areca catechu*, *Dictyosperma album*, *Roystonea regia*, *Eucalyptus grandis*, *Thysanolaena maxima*, *Avena sativa*, *Oryza sativa*, and *Sorghum bicolor* [5,10,11,13,14].

According to Schuster and Coyne [15,16], several plant-pathogenic bacteria can survive between crop cycles in annual and perennial plant hosts, which often do not show any symptoms. Bacterial survival in alternative host plants is an important source of inoculum for posterior crops. New information on the survival of *Xvv* in cultivated or spontaneous plant species is of paramount importance for appropriate management of this disease in maize-producing areas. The pathogenicity of *Xvv* in oats, rice, sugarcane, and sorghum has previously been experimentally demonstrated in the U.S. using artificial inoculation [5,11], but the potential of these cultivated plants and non-cultivated ones as important alternative hosts for *Xvv* has not yet been investigated in Brazil.

In this context, a wide range of annual mono-and dicotyledonous plants, including cultivated plants, cover crops, grasses, and forage that undergo succession and crop rotation with maize, and weeds with occurrence in maize-producing areas, were evaluated in this study in order to identify potential alternative hosts for *Xvv*. These plant species were inoculated with *Xvv* under greenhouse conditions. Accordingly, we investigated (i) the pathogenicity of *Xvv* for different plant species through visual evaluation of symptoms and re-isolation of the bacterium, (ii) epiphytic colonization of *Xvv* in asymptomatic plants, and (iii) bacterial colonization in plant tissues by scanning electron microscopy (SEM) in species that were symptomatic and asymptomatic.

## 2. Materials and Methods

### 2.1. Origin and Plant Cultivation

We included 52 species of plants in this study: 28 mono- and 24 dicotyledons (Tables 1–3). More than 1cultivar was tested for *Avena sativa*, *Brachiaria brizantha*, *B. decumbens*, and *Panicum maximum*, comprising 58 genotypes. The common maize hybrid IPR 164 and white maize hybrid IPR 127 were included as susceptible hosts to BLS [17]. Seeds of annually cultivated plants, cover crops, forage, and weeds were provided by the Instituto de Desenvolvimento Rural do Paraná—IAPAR/Emater (IDR-Paraná), Londrina, state of Paraná, Brazil. Grass seeds were provided by the Empresa Brasileira de Pesquisa Agropecuária (Embrapa—Gado de Corte), Campo Grande, state of Mato Grosso do Sul, Brazil.

**Table 1.** Pathogenicity and epiphytic colonization of *Xanthomonas vasicola* pv. *vasculorum* (*Xvv*), the causal agent of bacterial leaf streak (BLS) of maize, to cultivated plants and cover crops.

| Botanical Group | Scientific Name | Common Name | Cultivar | S[1] | EC[2] |
|---|---|---|---|---|---|
| Monocotyledonous Poaceae | | | | | |
| | *Avena sativa* | Oat | IPR Afrodite | + | N/A |
| | *A. sativa* | Oat | IPR Esmeralda | + | N/A |
| | *A. strigosa* | Black oat | IPR 161 | + | N/A |
| | *Hordeum vulgare* | Barley | BRS Cauê | + | N/A |
| | *Oryza sativa* | Rice | IPR 117 | + | N/A |
| | *Pennisetum glaucum* | Millet | IPR St. Tereza do Oeste | − | + |
| | *Saccharum officinarum* | Sugar cane | Not determined | − | + |
| | *Secale cereale* | Rye | IPR 89 | − | + |
| | *Sorghum bicolor* | Sorghum | BRS 658 | − | + |
| | ×*Triticosecale* | Triticale | IPR Caiapó | − | + |
| | *Triticum aestivum* | Wheat | IPR Potyporã | − | + |
| | *Zea mays*[3] | White maize | IPR 127 | + | N/A |
| | *Z. mays*[3] | Common maize | IPR 164 | + | N/A |

**Table 1.** *Cont.*

| Botanical Group | Scientific Name | Common Name | Cultivar | S[1] | EC[2] |
|---|---|---|---|---|---|
| Dicotyledonous | | | | | |
| Asteraceae | *Helianthus annuus* | Sunflower | BRS 417 | − | + |
| | *Glycine max* | Soybean | Potência | − | + |
| | *Lupinus albus* | White lupin | Not determined | − | + |
| | *L. angustifolius* | Blue lupin | IPR 124 | − | + |
| Fabaceae | *Mucuna aterrima cv.* Mucuna preta | Black velvet-bean | Not determined | − | + |
| | *M. pruriens* | Bengal velvet-bean | Not determined | − | + |
| | *Phaseolus vulgaris* | Bean | Carioca | − | + |
| Malvaceae | *Gossypium hirsutum* | Cotton | FMT 701 | − | + |
| Polygonaceae | *Fagopyrum esculentum* Moench | Buckwheat | IPR 91 | − | + |

S[1], symptoms were assessed at 15 DAI and *Xvv* association confirmed by PCR. EC[2], epiphytic colonization: *Xvv* isolation in asymptomatic plants was performed at 21 DAI in the first experiment and at 15, 21, and 30 DAI in the second experiment. The isolated *Xvv* was confirmed by PCR. +, presence of symptoms or *Xvv*; −, absence of symptoms or *Xvv*; N/A, not assessed. *Zea mays*[3], IPR 127 and IPR 164 maize hybrids were included in the experiment as *Xvv*-positive controls.

**Table 2.** Pathogenicity and epiphytic colonization of *Xanthomonas vasicola* pv. *vasculorum* (*Xvv*), the causal agent of bacterial leaf streak (BLS) of maize, in forage and grasses.

| Botanical Group | Scientific Name | Common Name | Cultivar | S[1] | EC[2] |
|---|---|---|---|---|---|
| Monocotyledonous | | | | | |
| Poaceae | *Brachiariabrizantha* | Brizantha bread grass | Not determined | + | N/A |
| | *B. brizantha* | Marandu bread grass | Marandu | + | N/A |
| | *B. brizantha* | Piatã bread grass | BRS Piatã | − | + |
| | *B. brizantha* | MG-5 Vitória bread grass | MG-5 Vitória | − | + |
| | *B. decumbens* | Basilisk bread grass | Basilisk | − | + |
| | *B. decumbens* | Decumbens bread grass | Not determined | − | + |
| | *B. humidicola* | Humidicola bread grass | Not determined | − | + |
| | *B. ruzisiensis* | Ruzisiensis bread grass | Not determined | − | + |
| | *Panicum maximum* | MG-12 Paredão Guinea grass | MG12 Paredão | − | + |
| | *P. maximum* | Tanzania Guinea grass | Tanzânia-1 | − | + |
| | *Zea mays*[3] | White maize | IPR 127 | + | N/A |
| | *Z. mays*[3] | Common maize | IPR 164 | + | N/A |
| Dicotyledonous | | | | | |
| Fabaceae | *Cajanus cajan* | Pigeon pea | IAPAR 43 | − | + |
| | *Crotalaria spectabilis* | Showy rattlepod | Not determined | − | + |

S[1], symptoms were assessed at 15 DAI and *Xvv* association confirmed by PCR. EC[2], epiphytic colonization: *Xvv* isolation in asymptomatic plants was performed at 21 DAI in the first experiment and at 15, 21, and 30 DAI in the second experiment. The isolated *Xvv* was confirmed by PCR. +, presence of symptoms or *Xvv*; −, absence of symptoms or *Xvv*; N/A, not assessed. *Zea mays*[3], IPR 127 and IPR 164 maize hybrids were included in the experiment as *Xvv*-positive controls.

Seeds of each plant species were sown in 8L plastic pots containing a soil/sand/organic matter mixture (3:1:1) free of BLS bacterium. The number of seeds per pot varied according to species and seed size. After seedling emergence, thinning was performed, leaving at least 10 seedlings per pot for grasses, forage, cover crops, andweeds, and 3–10 plants for cultivated plants based on size. For each genotype, 4 potted plants were used: 1 for the negative control and 3 containing plants inoculated with *Xvv* suspension. Potted plants were maintained in a semi-climatized greenhouse at the IDR-Paraná in Londrina, state of Paraná, Brazil, under partially controlled temperature and relative humidity (Figure 1).

**Table 3.** Pathogenicity and epiphytic colonization of *Xanthomonas vasicola* pv. *vasculorum* (*Xvv*), the causal agent of bacterial leaf streak (BLS) of maize, in weeds.

| Botanical Group | Scientific Name | Common Name | S[1] | EC[2] |
|---|---|---|---|---|
| Monocotyledonous | | | | |
| Poaceae | *Brachiaria plantaginea* | Plantain signalgrass | − | + |
| | *Cenchrus echinatus* | Southern sandbur | − | + |
| | *Chloris gayana* | Rhodes grass | − | + |
| | *C. polydactyla* | Finger grass | − | − |
| | *Digitaria horizontalis* | Jamaican crabgrass | + | N/A |
| | *D. insularis* | Sourgrass | + | N/A |
| | *Echinochloa colonum* | Jungle rice | + | N/A |
| | *Eleusine indica* | Crowfootgrass | + | N/A |
| | *Lolium multiflorum* | Annual ryegrass | − | + |
| | *Panicum maximum* | Guinea grass | − | + |
| | *Pennisetum purpureum* | Napier grass | − | + |
| | *Rhynchelytrum repens* | Natal grass | − | + |
| | *Sorghum arundinaceum* | Common wild sorghum | + | N/A |
| | *Zea mays*[3] | White maize | + | N/A |
| | *Z. mays*[3] | Common maize | + | N/A |
| Dicotyledonous | | | | |
| Amaranthaceae | *Amaranthus hybridus* | Smooth pigweed | − | + |
| | *A. viridis* | Slender amaranth | − | + |
| Asteraceae | *Acanthospermum hispidum* | Bristly starbur | − | + |
| | *Bidens pilosa* | Beggar's tick | − | + |
| | *Conyza* spp. | Hairy fleabane | − | + |
| | *Galinsoga parviflora* | Gallant soldier | − | − |
| | *Tridax procumbens* | Coatbuttons | − | + |
| Brassicaceae | *Raphanus raphanistrum* | Wild radish | − | + |
| | *R. sativus* | Radish | − | + |
| Commelinaceae | *Commelina benghalensis* | Wandering jew | − | + |
| Euphorbiaceae | *Euphorbia heterophylla* | Wild poinsettia | − | + |
| Malvaceae | *Sidarhom bifolia* | Arrow-leaf sida | − | + |
| Rubiaceae | *Richardia brasiliensis* | Tropical Mexican clover | − | − |

S[1], symptoms were assessed at 15 DAI and *Xvv* association confirmed by PCR. EC[2], epiphytic colonization: *Xvv* isolation in asymptomatic plants was performed at 21 DAI in the first experiment and at 15, 21, and 30 DAI in the second experiment. The isolated *Xvv* was confirmed by PCR. +, presence of symptoms or *Xvv*; −, absence of symptoms or *Xvv*; N/A, not assessed. *Zea mays*[3], IPR 127 and IPR 164 maize hybrids were included in the experiment as *Xvv*-positive controls.

### 2.2. Isolates of Xvv and Plant Inoculation

The RL1 strain of *Xvv* [1] from the bacterial culture collection of the Bacteriology Laboratoryof the IDR-Paraná, Londrina, state of Paraná, Brazil, was used in this study for plant inoculation. The identity of the bacterial culture was confirmed by polymerase chain reaction (PCR), using the specific primers *Xvv*3_F (5′-CAAGCAGAGCATGGCAAAC-3′) and *Xvv*3_R (5′-CACGTAGAACCGGTCTTTTGG-3′) specific for *Xvv* as reported by Lang et al. [11]. This primer set amplifies a 207-bp fragment of the bacterial genome [11]. The pathogenicity of the bacterial isolates was confirmed by inoculating the hybrid maize cv. IPR 164 maintained under controlled conditions.

The RL1 strain suspensions used for plant inoculations were prepared from bacterial cultures grown on nutrient-agar (NA) medium at 28 °C for 24–72 h. The aerial parts of the plants were artificially inoculated with a bacterial suspension at a concentration of $1 \times 10^8$ CFU·mL$^{-1}$ of the RL1 strain of *Xvv* (OD$_{600}$ = 0.1, Genesys™ 150, ThermoFisher Scientific, Madison, WI, USA). Plants were inoculated by spraying until complete runoff (~20–30 mL per plant). Control plants were sprayed with sterilized distilled water. All plants were maintained in a humid chamber for 24 h before and after inoculation. Maize plants were inoculated at the V3–V4 phenological stage, while rice, oat, millet, wheat, triticale, buckwheat, and radish plants were inoculated 17 days after sowing when plants

had 5 or more expanded leaves. *Crotalaria spectabilis*, pigeon pea, white lupin, and other plants were inoculated 30 days after sowing. The experiments were repeated twice.

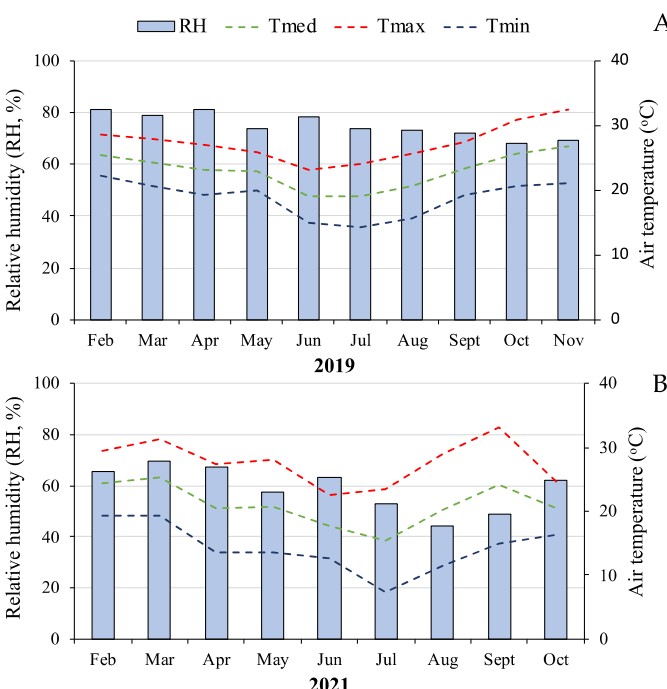

**Figure 1.** Maximum (Tmax), minimum (Tmin), and average (Tmed) air temperature (°C) and relative humidity (RH, %) inside a semi-climatized greenhouse at the Instituto de Desenvolvimento Rural do Paraná-IAPAR/Emater, Londrina, PR, Brazil, from February to November 2019 (**A**) and February to October 2021 (**B**).

### 2.3. Pathogenicity and Epiphytic Colonization of Xvv in Multiple Plant Species

The pathogenicity of *Xvv* in different plants was determined by the development of symptoms and bacterial re-isolation according to Lang et al. [11]. Fifteen days after inoculation (DAI), plants were visually evaluated for the presence of symptoms on leaves. Once symptoms were observed, symptomatic leaves were collected, taken to the laboratory, and subjected to re-isolation of *Xvv*. Leaf tissues between healthy and diseased areas were cut in small sections (2–4 mm), placed in 70% alcohol for 1 min, transferred to 1.0% sodium hypochlorite (NaOCl) for 1 min, and washed 3 times in sterilized distilled water. After surface disinfection, the leaf tissues were macerated in sterilized water. After 5 min, aliquots were deposited in Petri dishes containing NA medium (Kasvi®, São José dos Pinhais, PR, Brazil) and scattered using a Drigalski spreader. After 24–72 h, plates were examined for the presence of *Xvv* colonies. The identity of the bacterial colonies was confirmed by PCR, as described in Section 2.4. Plants were considered symptomatic when showing typical symptoms of the bacterial infection, and confirmed by PCR assay. Plants that did not show any visual symptoms of *Xvv* infection were subjected to isolation of the epiphytic bacterial population. The evaluation of the epiphytic bacterial population inthe leaf samples was performed 21 DAI in the first experiment. In the second experiment, this evaluation was carried out at 15, 21, and 30 DAI.

To evaluate the epiphytic population, 1.0 g of leaves was placed in 50 mL phosphate-buffered saline (PBS: 6.25 g $KH_2PO_4$, 8.75 g $K_2HPO_4$, 1.0 g bacteriological peptone, 1.0 L distilled water, and 1 aliquot of Tween 20) [18], in a 125 mL Erlenmeyer flask and stirred at 150 rpm and 28 °C for 1 h using an orbital shaker. After incubation, leaf washes were diluted to $10^{-2}$ in PBS buffer and placed in Petri dishes containing NA medium supplemented with cyclohexamide (50 mg·mL$^{-1}$). Plates were maintained at 28 °C and evaluated after 24–72 h. The epiphytic bacterial population was determined by counting the colonies that grew on the medium, which were expressed as log CFU·g$^{-1}$. Five to ten colonies per

plate, suspected to be *Xvv*, were selected to confirm *Xvv* identity by PCR, according to the methodology described in the following section.

### 2.4. PCR Assessment for Xvv

The total genomic DNA of each bacterial culture was obtained from a suspension of $1 \times 10^8$ CFU·mL$^{-1}$ (1.0 mL) subjected to a 100 °C water bath for 5 min, followed by rapid cooling in ice [19]. The PCR reaction was performed using specific oligonucleotide primers for *Xvv* (*Xvv*3_F and *Xvv*3_R) [11]. The final volume of the reaction was 25 μL, which included a mixture of 1.0 μL of each specific primer (0.5 mM), 1.0 μL of dNTP (5 mM), 0.8 μL of MgCl$_2$ (50 mM), 2.5 μL of buffer (1×), 0.2 μL of Recombinant *Taq* DNA Polymerase (5 U·μL$^{-1}$), 17.5 μL of ultrapure water, and 1.0 μL of total DNA. Thirty cycles of amplification were performed, comprising denaturing at 94 °C for 30 s, primer annealing at 55 °C for 30 s, and final extension at 72 °C for 1 min in a Verit 96-well thermal cycler (Applied Biosystems™, Marsiling, Woodlands, Singapore).

The PCR products were electrophoresed on a 1.0% (*w/v*) agarose gel in 1× TAE buffer. Samples of *Xvv*-inoculated maize plants were used as positive controls. Agarose gel was pre-stained with ethidium bromide (0.1 μg·mL$^{-1}$). The PCR products were visualized and photographed (L-PIX EX, Loccus do Brasil Ltd., Cotia, SP, Brazil) under UV light.

### 2.5. Scanning Electron Microscopy (SEM) Analysis of Plants Colonized with Xvv

Symptomatic leaf tissues of oat cvs. IPR Afrodite and IPR Esmeralda, black oat cv. IPR 161, maize cv. IPR 164, rice cv. IPR 117, millet cv. IPR Santa Teresa do Oeste, sorghum cv. BRS 658, and rye cv. IPR 89 were cut with a 0.5 mm diameter awl. Leaf samples were prepared for SEM and examined at the Electron Microscopy and Microanalysis Laboratory (LMEM) of the Universidade Estadual de Londrina (UEL), Londrina, PR, Brazil. Plant material was fixed in a 2.5% glutaraldehyde + 3.0% paraformaldehyde + 0.2 M sodium cacodylate buffer (pH 7.2) for 4 h, followed by washing in 0.1 M sodium cacodylate buffer ($3 \times 10$ min each) and post-fixing with 1.0% osmium tetroxide (OsO$_4$) for 1 h. Samples were washed again in 0.1 M sodium cacodylate buffer ($3 \times 10$ min each) and dehydrated in increasing concentrations of ethanol (50%, 60%, 70%, 80%, 90%, and 100%) for 10 min each, except for the 100% concentration that was subjected to two washes of 10 min each. Afterwards, the specimens were subjected to critical point drying with liquid CO$_2$ (Critical Point Dryer CPD 030, Bal-Tec, Balzers, Liechtenstein). They were then placed on stubs containing double-sided carbon adhesive tape, placed on a metalizer (Sputter Coater—SCD 050, Bal-Tec, Balzers, Liechtenstein) to cover the surface with a thin layer of gold, and examined under a scanning electron microscope (Quanta 200, Philips FEI, Eindhoven, Netherlands) [20]. Some samples were subjected to brittle fracture by cutting with a scalpel before SEM analysis.

## 3. Results

### 3.1. Pathogenicity and Epiphytic Colonization of Different Plant Species by Xvv

Among the 58 plant species and cultivars evaluated in this study, 12 genotypes were symptomatic when inoculated with the RL1 strain of *Xvv* (Tables 1–3). The symptomatic species were *Avena sativa* (cv. IPR Afrodite and cv. IPR Esmeralda), *A. strigosa* (cv. IPR 161), *Hordeum vulgare* (cv. BRS Cauê), *Oryza sativa* (cv. IPR 117), *Brachiaria brizantha* (Brizantha and cv. Marandu), *Digitaria horizontalis*, *D. insularis*, *Echinochloa colonum*, *Eleusine indica*, and *Sorghum arundinaceum*. All species belonged to the same botanical family, Poaceae, and can be potential hosts of *Xvv* (Tables 1–3). None of the other species showed any typical symptoms when inoculated with the RL1 strain of *Xvv* (Tables 1–3). However, *Xvv* was detected epiphytically colonizing the phyllosphere of most cultivated plants, cover crops, forage, grasses, and weeds included in this study (Tables 1–3). Among the weeds, only three species, *Chloris polydactyla* (Poaceae), *Galinsoga paviflora* (Asteraceae), and *Richardia brasiliensis* (Rubiaceae), showed no epiphytic colonization by *Xvv* (Table 3).

The epiphytic survival period of the RL1 strain varied among the plant species (Table 4). For the cultivated plants and cover crop groups, *Xvv* showed epiphytic survival for up to 30 days on millet, rye, wheat, and bean, while sugarcane, triticale, sunflower, white and blue lupin, velvet bean, cotton, and buckwheat had survival periods of 21 days (Table 4). Regarding forage and grass species, *Xvv* survival was up to 30 days in *B. decumbens* (cv. Basilisk), and *Crotalaria spectabilis*, and up to 21 days in *B. brizantha* (cv. Piatã and MG-5 Vitória), *B. decumbens*, *B. humidicola*, *B. ruzisiensis*, *Panicum maximum* cv. MG-12 Paredão, and *Cajanus cajan* (Table 4). *Xvv* survived for only 15 days on *P. maximum* cv. Tanzânia (Table 4).

**Table 4.** Maximum recovery period in days after inoculation (DAI) and colony-forming units (Log$_{10}$CFU·g$^{-1}$) of *Xanthomonas vasicola* pv. *vasculorum* (*Xvv*), the causal agent of bacterial leaf streak (BLS) of maize, in cultivated plants, cover crops, forage, grasses and weeds.

| Botanical Group | Scientific Name | Common Name | *Xvv* Maximum Recovery Period (DAI) [1] | CFU (Log$_{10}$·g$^{-1}$) [2] |
|---|---|---|---|---|
| Cultivated plants and cover crops | | | | |
| Poaceae | *Pennisetum glaucum* | Millet | 30 | 5.46 |
| | *Saccharum officinarum* | Sugarcane | 21 | 4.40 |
| | *Secale cereale* | Rye | 30 | 6.36 |
| | *Sorghum bicolor* | Sorghum | 21 | 5.18 |
| | ×*Triticosecale* | Triticale | 21 | 5.40 |
| | *Triticum aestivum* | Wheat | 30 | 6.71 |
| Asteraceae | *Glycine max* | Soybean | 21 | 3.30 |
| | *Helianthus annuus* | Sunflower | 21 | 6.02 |
| | *Lupinus albus* | White lupin | 21 | 7.00 |
| | *L. angustifolius* | Blue lupin | 21 | 4.00 |
| Fabaceae | *Mucuna pruriens* | Bengal velvet-bean | 21 | 4.30 |
| | *M. aterrima* cv. Mucuna preta | Black velvet-bean | 21 | 6.01 |
| | *Phaseolus vulgaris* | Bean | 30 | 5.00 |
| Malvaceae | *Gossypium hirsutum* | Cotton | 21 | 5.04 |
| Polygonaceae | *Fagopyrum esculentum* Moench | Buckwheat | 21 | 5.93 |
| Forages and grasses | | | | |
| Poaceae | *Brachiaria brizantha* | Piatã bread grass | 21 | 3.48 |
| | *B. brizantha* | MG-5 Vitória bread grass | 21 | 3.65 |
| | *B. decumbens* | Decumbensbread grass | 21 | 4.74 |
| | *B. decumbens* | Basilisk bread grass | 30 | 2.70 |
| | *B. humidicola* | Humidicola bread grass | 21 | 4.00 |
| | *B. ruzisiensis* | Ruzisiensis bread grass | 21 | 5.00 |
| | *Panicum maximum* | Tanzânia Guinea grass | 15 | 4.00 |
| | *P. maximum* | MG-12 Paredã Guinea grass | 21 | 4.81 |
| Fabaceae | *Cajanus cajan* | Pigeon pea | 21 | 4.18 |
| | *Crotalaria spectabilis* | Showy rattlepod | 30 | 5.23 |
| Weeds | | | | |
| Poaceae | *B. plantaginea* | Plantain signalgrass | 21 | 5.95 |
| | *Cenchrus echinatus* | Southern sandbur | 30 | 5.00 |
| | *Chloris gayana* | Rhodes grass | 30 | 6.27 |
| | *Lolium multiflorum* | Annual ryegrass | 21 | 7.00 |
| | *P. maximum* | Guinea grass | 30 | 5.18 |
| | *Pennisetum purpureum* | Napier grass | 30 | 5.92 |
| | *Rhynchelytrum repens* | Natal grass | 30 | 6.40 |
| Amaranthaceae | *Amaranthus hybridus* | Smooth pigweed | 30 | 3.85 |
| | *A. viridis* | Slender amaranth | 30 | 3.98 |
| Asteraceae | *Acanthospermum hispidum* | Bristly starbur | 30 | 5.18 |
| | *Bidens pilosa* | Beggar's tick | 21 | 5.00 |
| | *Conyza* spp. | Hairy fleabane | 15 | 4.00 |
| | *Tridax procumbens* | Coatbuttons | 30 | 3.70 |
| Brassicaceae | *Raphanus raphanistrum* | Wild radish | 30 | 5.48 |
| | *Raphanus sativus* | Radish | 15 | 4.28 |
| Commelinaceae | *Commelina benghalensis* | Wandering jew | 30 | 5.18 |
| Euphorbiaceae | *Euphorbia heterophylla* | Wild poinsettia | 30 | 3.93 |
| Malvaceae | *Sidarhombifolia* | Arrow-leaf sida | 30 | 5.00 |

[1], DAI, days after inoculation. [2], Maximum value of epiphytic population recovered from the phyllosphere of plants in the two experiments, considering all the evaluations.



*Xvv* survived epiphytically for up to 30 days on most of the weeds tested (Table 4). In *Lolium multiflorum* (Poaceae) and *Bidens pilosa* (Asteraceae), bacterial survival was 21 days, whereas in *Conyza* spp. (Asteraceae) and *Raphanus sativus* (Brassicaceae), the epiphytic survival period was only 15 days (Table 4). Regarding the epiphytic colonization of *Xvv*, differences were observed among the species, ranging from 2.70 $\log_{10}$ CFU·g$^{-1}$ of leaf tissue in *B. decumbens* cv. Basilisk at 30 DAI, to 7.00 $\log_{10}$ CFU·g$^{-1}$ in *L. multiflorum* and *L. albus* at 21 DAI (Table 4). Results also indicated variation in the epiphytic populations of *Xvv* in different leaves of the same plant.

The results obtained in this study also indicate that several non-host plant species can favor epiphytic colonization by *Xvv*. These plants could be a source of *Xvv* inoculum for BLS outbreaks in maize-producing areas. Therefore, knowledge of potential symptomatic and asymptomatic host plants that can serve as *Xvv* sources is of paramount importance for epidemiological studies and the establishment of measures to better manage this disease.

In symptomatic species, foliar symptoms were observed from seven up to nine days after inoculation, including small circular or irregular lesions of soaked tissue, typical of anasarca, which were well-defined on both sides of the leaves. After this period, differences were observed in symptoms among the species (Figures 2 and 3). In maize, lesions developed, but they remained restricted to the interveinal regions of the leaf and had a translucent appearance when visualized against light. Lesions on the other species of cultivated plants, cover crops, forage, and grasses, had variations in shape and color (Figures 2 and 3).

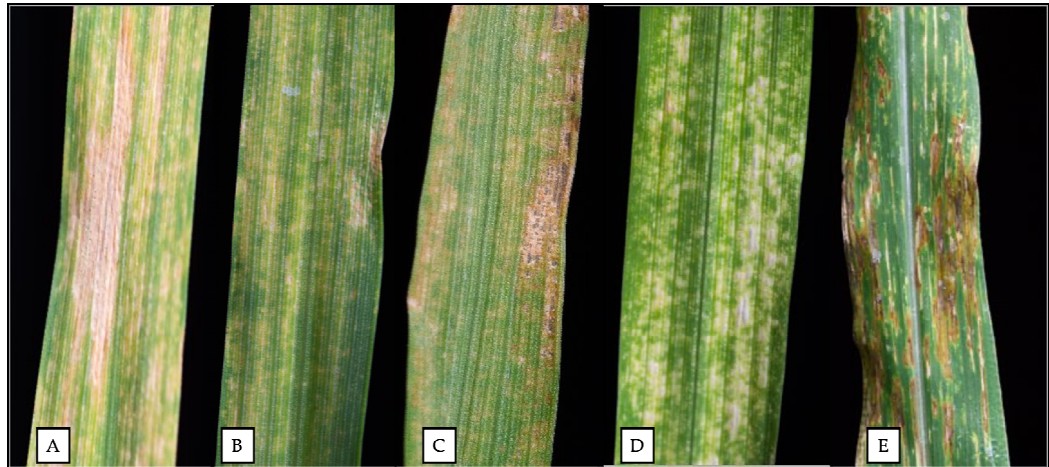

**Figure 2.** Leaves of cultivated plants showing symptoms of bacterial leaf streak (BLS) 15 days after inoculation (DAI) with the RL1 strain of *Xanthomonas vasicola* pv. *vasculorum* (*Xvv*): oat cvs. IPR Afrodite (**A**) and IPR Esmeralda (**B**) with yellowed spots on leaves progressing to chlorosis; (**C**) black oat cv. IPR 161 with yellowed spots on leaves progressing to chlorosis; (**D**) barley cv. BRS Cauê with straw-yellow spots along the veins; and (**E**) maize cv. IPR 164 with streaks and wavy margins on leaves.

In oat, the lesions, initially brownish-yellow in color, were numerous and coalescent, covering a substantial part of the leaf surface. Some areas also became chlorotic (Figure 2A–C). Several lesions were evident on oat, but the severity of the symptoms was not evaluated. The oat cv. IPR Afrodite had the highest number of lesions. In barley, the symptoms presented as multiple straw-yellow to cream-colored punctuations, with the occurrence of lesion coalescence (Figure 2D). Common maize, used as *Xvv*-positive controls, showed typical lesions of BLS in the form of yellow to brown streaks along the interveinal regions, progressing to necrosis (Figure 2E). Bacterial exudates were observed on the adaxial surface of the leaf, indicating extensive multiplication of *Xvv* in the tissues.

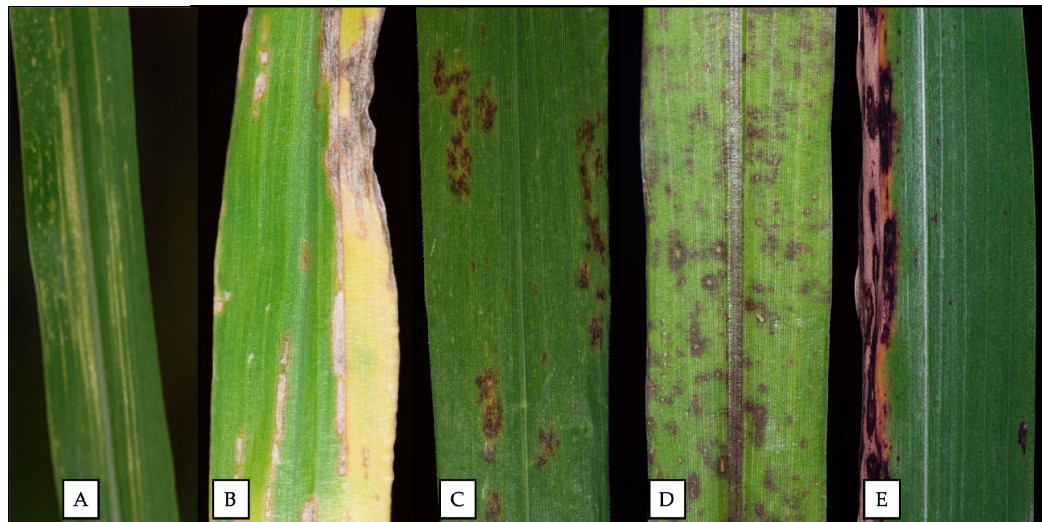

**Figure 3.** Leaves of forages and weeds showing symptoms of bacterial leaf streak (BLS) 15 days after inoculation (DAI) with the RL1 strain of *Xanthomonas vasicola* pv. *vasculorum* (*Xvv*): (**A**) *Brachiaria brizantha* (Brizantha) with symptoms of BLS; (**B**) *Digitaria insularis* with symptoms of chlorosis; (**C**) *Echinochloa colonum* with red spots; (**D**) *Digitaria horizontalis* with reddish punctuations; and (**E**) *Sorghum arundinaceum* with red spots on the leaf surface and necrosis in the leaf margin.

Lesions on leaves of *B. brizantha* (Brizantha) were characterized by very narrow streaks (1–2 mm wide) with well-defined margins, which were brownish-yellow in color and parallel to the veins (Figure 3A). Considering all tested weeds, *D. insularis* had the most severe symptoms, consisting of brownish-yellow streaks surrounded by a yellow leaf area, covering a large surface of the leaf blade (Figure 3B). In *E. colonum*, the lesions consisted of small, round, reddish spots that were irregularly distributed along the leaf (Figure 3C). In *D. horizontalis*, reddish lesions were distributed throughout the leaf (Figure 3D). In *S. arundinaceum*, the lesions were bright, wine-red in color, and coalesced, forming large irregular spots (Figure 3E). Regarding the severity of symptoms in this group of symptomatic forage plants and weeds, it was possible to observe evident differences in the severity of the disease, with *D. insularis* showing the most severe symptoms, although this was not quantitatively evaluated. Bacteria were isolated from symptomatic leaves to obtain pure cultures of *Xvv*. The identity of the cultures was confirmed by PCR.

*3.2. Scanning Electron Microscopy (SEM) Analysis of Plants Colonized by Xvv*

Scanning electron microscopy (SEM) was performed on plant samples from symptomatic species (oat cvs. IPR Afrodite and IPR Esmeralda, black oat cv. IPR 161, maize cv. IPR 164, and rice cv. IPR 117) and on three other species of the Poaceae family, which did not show any typical disease symptoms (sorghum cv. BRS 658, rye cv. IPR 89, and millet cv. IPR St. Tereza do Oeste) when inoculated with the RL1 strain of *Xvv*. Samples were collected at 15 DAI, when symptomatic plants showed typical symptoms due to infection by the bacterium.

*Xvv* cells and aggregates were frequently observed on the adaxial surface of the leaves of oat cvs. IPR Afrodite and IPR Esmeralda (Figure 4A,C), black oat (Figure 4E), maize (Figure 5A), rice (Figure 5C), sorghum (Figure 6A), and rye (Figure 6B). Fracture images revealed bacterial colonization in the intercellular spaces and substomatal chambers in the leaf tissues of oat cvs. IPR Afrodite (Figure 4B) and IPR Esmeralda (Figure 4D), black oat cv. IPR 161 (Figure 4F), and maize IPR 164 (Figure 5B). Exopolysaccharides (EPS) and mucus were also observed around the bacterial cells (Figures 4B,D,F and 5B).

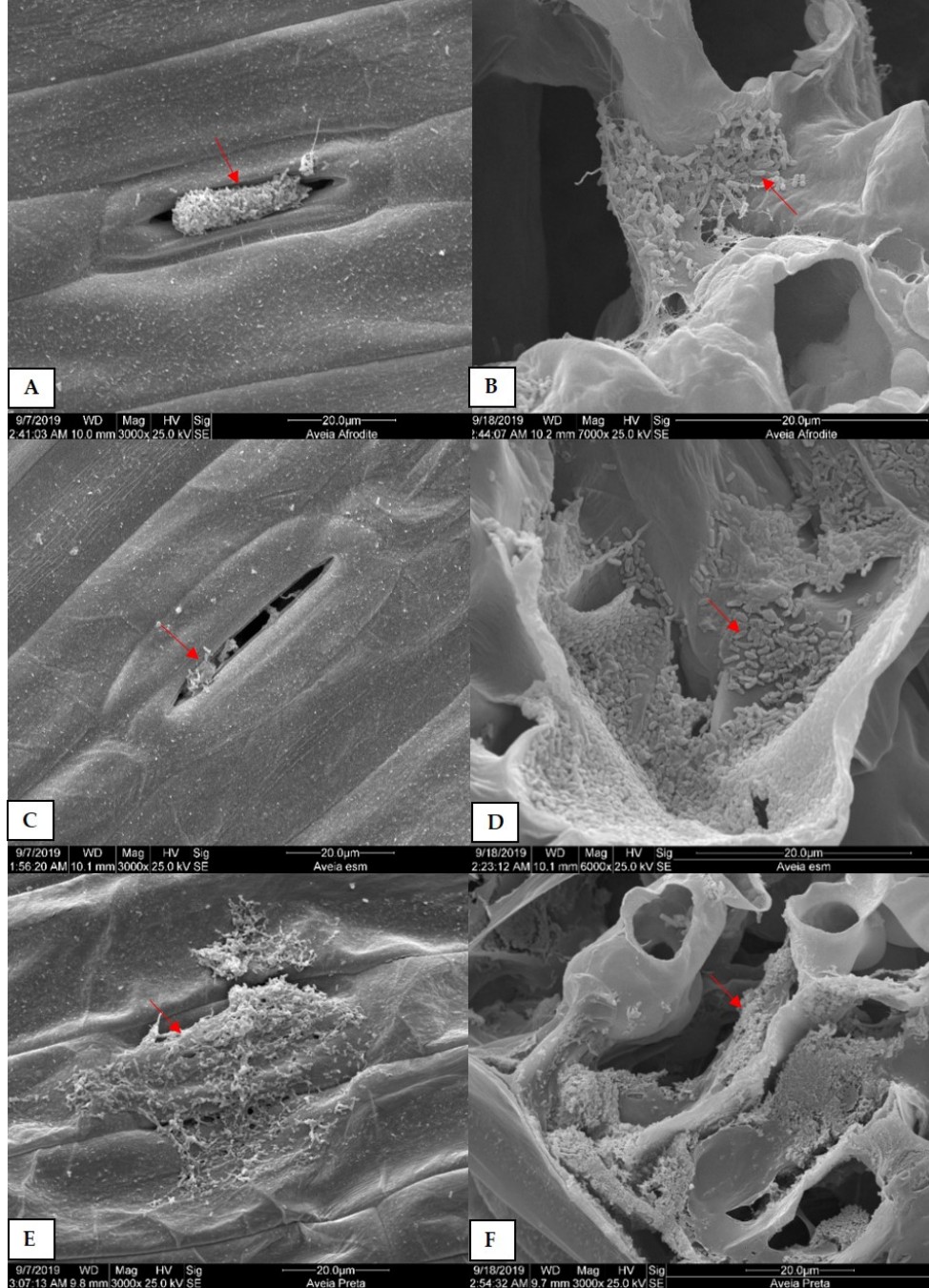

**Figure 4.** Scanning electron microscopy (SEM) of the leaf surface and leaf fractures of oat 15 days after inoculation (DAI) with *Xanthomonas vasicola* pv.*vasculorum* (*Xvv*). (**A**) Oat cv. IPR Afrodite with cluster of *Xvv* cells in the stomata (red arrow); (**B**) oat cv. IPR Afrodite with *Xvv* colonization in the leaf mesophyll with bacterial aggregates and production of amorphous material suggesting exopolysaccharides (EPS, red arrow); (**C**) oat cv. IPR Esmeralda with cluster of *Xvv* cells in the stomata (red arrow); (**D**) oat cv. IPR Esmeralda with *Xvv* colonization in the leaf mesophyll with bacterial aggregates and production of amorphous material suggesting EPS (red arrow); (**E**) black oat cv. IPR161 with cluster of *Xvv* cells in the stomata (red arrow); (**F**) black oat cv. IPR161 with *Xvv* colonization in leaf mesophyll with bacterial aggregates and production of amorphous material suggesting EPS (red arrow).

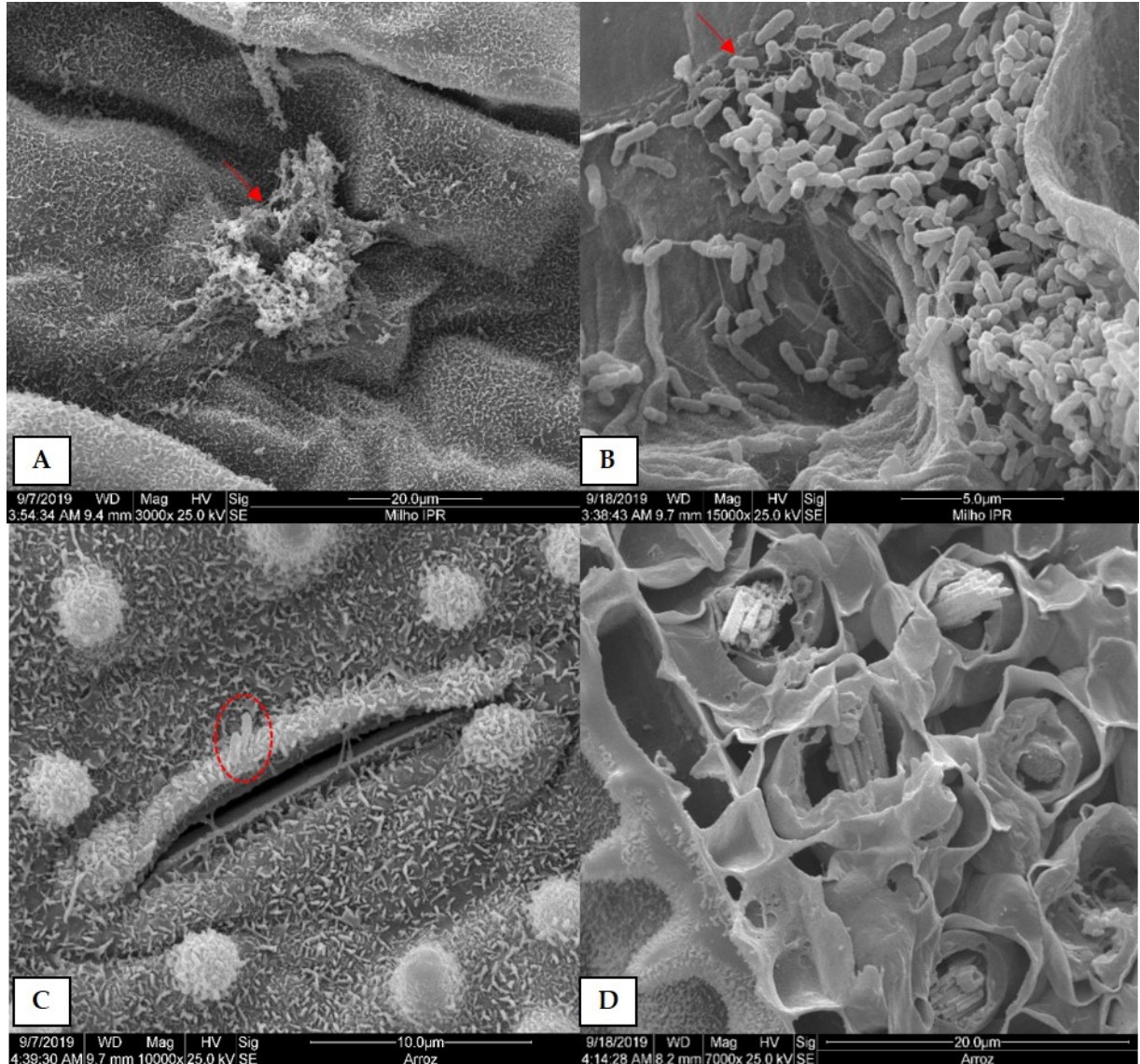

**Figure 5.** Scanning electron microscopy (SEM) of the adaxial surface of leaves and leaf fractures of maize cv. IPR 164 and rice cv. IPR 117 at 15 days after inoculation (DAI) with *Xanthomonas vasicola* pv. *vasculorum* (*Xvv*). (**A**) Maize cv. IPR 164 with cluster of *Xvv* cells in the stomatal chamber (red arrow); (**B**) maize IPR 164 with cluster of *Xvv* cells in the leaf mesophyll covered by amorphous material suggesting exopolysaccharides (EPS, red arrow); (**C**) rice cv. IPR 117 with *Xvv* cells colonizing the region close to the stomata (dashed ellipse); (**D**) rice cv. IPR 117 with internal structure of the leaf mesophyll, without evidence of bacterial cells.

In rice, the presence of bacterial cells around the stomata was detected (Figure 5C); however, it was not possible to visualize the colonization of *Xvv* in the intercellular spaces in the fractures (Figure 5D). In sorghum (Figure 6A) and rye (Figure 6B), even when asymptomatic, it was possible to observe slight colonization of *Xvv* on leaf surfaces, in agreement with the results of the epiphytic survival of the bacteria verified for these species (Table 1). In millet, the morphology of bacterial cells was altered (Figure 6C), indicating possible cell death. Bacterial colonization of the mesophyll in these tissues was not observed during fracture examinations (Figure 6D). This suggests that millet is the only evaluated species on which *Xvv* can survive epiphytically (Table 1).

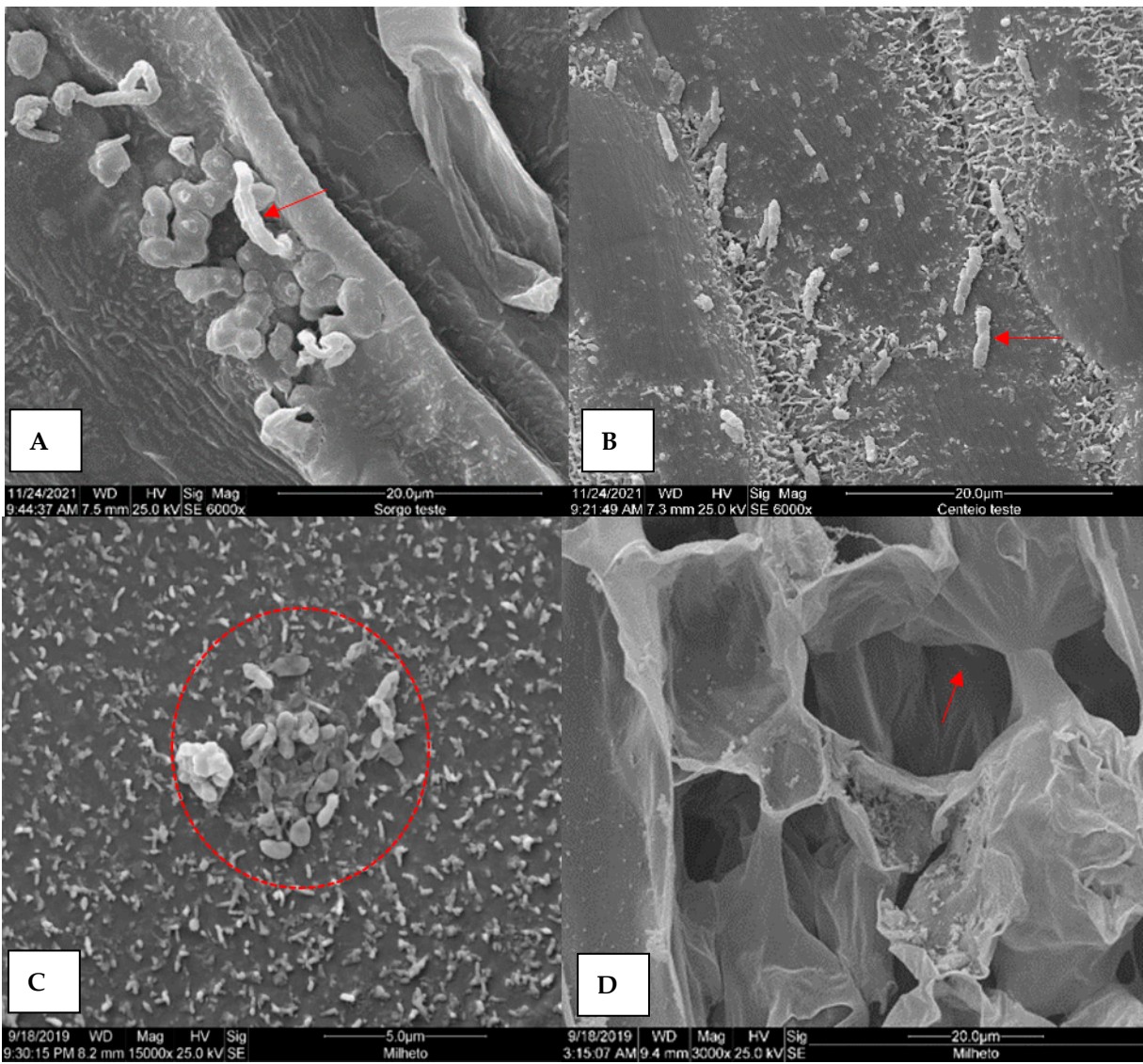

**Figure 6.** Scanning electron microscopy (SEM) of the surface of asymptomatic plants of sorghum cv. BRS 658, rye cv. IPR 89, and millet cv. IPR St. Tereza do Oeste at 15 days after inoculation (DAI) with *Xanthomonas vasicola* pv. *vasculorum* (*Xvv*). (**A**) Adaxial leaf surface of sorghum cv. BRS 658 with scattered *Xvv* cells in the epidermis (red arrow); (**B**) adaxial leaf surface of rye cv. IPR 89 with scattered *Xvv* cells in the epidermis (red arrow); (**C**) *Xvv* cell aggregates demonstrating altered morphology, indicating cell death on the adaxial surface of millet cv. IPR St. Tereza do Oeste (dashed circle); (**D**) fractures showing the internal structure of the leaf mesophyll, without evidence of bacterial colonization in millet cv. IPR St. Tereza do Oeste (red arrow).

## 4. Discussion

*Xvv* pathogenicity was evaluated for 52 mono- and dicotyledonous plant species of different botanical groups. Ten species were identified as potentially symptomatic host plants for *Xvv*. These species were restricted to the Poaceae family, except *Cyperus esculentus*, which is a member of the Cyperaceae family. These results corroborate those of previous studies on alternative hosts for *Xvv* [5,11,14].

The results obtained in this study expand the range of potential host plants for *Xvv*, which previously included only a few species such as maize, sorghum, sugarcane, broom bamboo, palm species, and eucalyptus [8–11,13]. Although our findings demonstrate the pathogenicity of *Xvv* through artificial inoculation under controlled conditions for several plant species, including some not yet reported in the literature as hosts for this

bacterium, the results may not represent the natural pathogen–host interaction under field conditions. Artificial inoculations may favor pathogen infection, as the high concentration of the inoculum can act more aggressively than under natural conditions.

In a previous study [14], eight species identified as symptomatic hosts for *Xvv* under greenhouse conditions were transplanted and evaluated under natural field infestation of *Xvv*. However, only two of the eight species were infected and confirmed to be susceptible to *Xvv* under natural conditions of the disease occurrence. Moreover, it was observed that the symptoms in maize were more severe than those in the potential alternative hosts. Similar findings were reported by Lang et al. [11] and Hartman et al. [14], indicating that *Xvv* may have shown higher aggressiveness or adaptation to infect maize.

Sugarcane and sorghum have previously been reported as symptomatic hosts of *Xvv* in Nebraska and Kansas, in theU.S. [5,11]. Sorghum readily showed symptoms when artificially inoculated with *Xvv* under greenhouse conditions [11]. However, no symptoms were evident under field conditions, even in sorghum that was adjacent to BLS-affected maize. Interestingly, sugarcane and sorghum showed no typical symptoms when inoculated with the RL1 strain of *Xvv* in our study. However, it was possible to recover the bacterium from the phyllosphere of these plants up to 21 days after inoculation, indicating epiphytic colonization of these plants by *Xvv*.

The differences in the host range among studies can be partially explained by the pathogenic variability or genetic diversity of the *Xvv* strain, which was extensively studied in previous work [21,22]. These studies demonstrated that the aggressiveness level of *Xvv* varied among strains, even though all tested strains showed some level of pathogenicity to different sugarcane cultivars [23]. To date, comparative studies of *Xvv* strains from Brazil with strains from other countries have not been reported. However, genomic sequencing studies of different *Xvv* strains obtained in the U.S., Argentina, and South Africa revealed that these isolates belong to the same taxonomic clade [24]. However, it is still unclear whether the U.S. and Argentine *Xvv* strains differ from those from South Africa in terms of host plants [11]. Initial studies with one U.S. and one South African strain of *Xvv* and four strains of *X. vasicola* pv. *holcicola* showed no differences in symptoms in maize [11], although these studies were limited to few strains of *Xvv* and only one maize cultivar.

Similar to our findings, *A. sativa* and *O. sativa* have been reported as hosts of *Xvv* by Lang et al. [11] and Hartman et al. [14]. Three *Avena* spp. cultivars were evaluated in our study: two of *A. sativa* and one of *A. strigosa*, and different levels of *Xvv* susceptibility were observed. *A. sativa* cv. IPR Afrodite showed the most severe disease symptoms compared with the other cultivars. These results reinforce the importance of evaluating the reaction of different genotypes of each species to precisely and safely determine the susceptibility to *Xvv* infection. *H. vulgare* plants were symptomatic when inoculated with the RL1 strain of *Xvv*. This species was reported as a non-host for *Xvv* in an earlier study [14], as no symptoms or endophytic colonization were found at that time.

Furthermore, this is the first report of *Xvv* pathogenicity in Brazil for some speciesthat include *B. brizantha*, *B. brizantha* cv. Marandu, *D. horizontalis*, *D. insularis*, *E. colonum*, *E. indica*, and *S. arundinaceum*. Therefore, these results are important, as forage plants such as the *Brachiaria* group are perennial and are used in intercropping with maize. Simultaneous cultivation of the grains and straw of maize and the roughage of forage such as *Brachiaria* spp. are common for cattle grazing [25].

All evaluated weed species are common in Brazil, and inhabit major crop-producing areas, pastures, and gardens. These plant species have high reproductive capability [26]. Among them, *S. arundinaceum* and *D. insularis* are perennial species and are difficult to control.

The tropical and subtropical climate of Brazil favors some regions to have two or more crop seasons during the year in the same area, mainly soybean and maize off-season succession [27]. Based on recent crop seasons, three-quarters of the Brazilian maize production was from off-season maize, cultivated during the autumn and winter [28]. However, the investment in weed control for off-season maize is generally low, which results in

increased weed infestation in the main production areas [27]. *D. insularis* is considered one of the most problematic weeds for this continuous intensive system of crop production, particularly with the occurrence of herbicide resistance [29]. The severity of symptoms and epiphytic colonization of *Xvv* found in *D. insularis* indicates that this species is a potential host for bacterial survival in maize-producing areas. In this context, it is of utmost importance to mitigate strategies to control the BLS disease in areas of natural occurrence of the pathogen, considering all plant species that are present in the agricultural system of the region. Perennial hosts such as *B. decumbens* may represent a potential risk for maize production, as *Xvv* can persist in these hosts season by season [14]. Furthermore, the pathogen can survive in aboveground debris of the alternative hosts and might serve as natural reservoirs of inoculum for new infections. Therefore, further studies are required to fully understand the interactions between *Xvv* and potential host weed plants.

Bacterial survival on the surface and/or inside of cells of cultivated plants, forage, or weeds is one of the main sources of inoculum for the development of new bacterial disease outbreaks in the main production areas [15]. In the present study, *Xvv* was detected epiphytically colonizing several cultivated plants and cover crops, including mono- and dicotyledonous species. Thus, this bacterium can epiphytically colonize plants of different botanical groups, such as Amaranthaceae, Asteraceae, Brassicaceae, Commelinaceae, Euphorbiaceae, Fabaceae, Malvaceae, Poaceae, and Polygonaceae.

Our findings demonstrated a variation in the epiphytic populations sampled from different leaves of the same plant, which is commonly observed in epiphytic bacteria [30,31]. The population size of epiphytic bacteria can vary by more than 1000 times from one leaf to another, even when grown under similar environmental conditions [31–34].

When analyzing the epiphytic colonization of *Xvv* in forage, we found that all tested species enabled bacterial survival. Regarding weeds, 86% of the tested species showed epiphytic survival of the pathogen. In contrast, *C. polydactyla*, *G. paviflora*, and *R. brasiliensis* showed no epiphytic colonization by *Xvv*. This could be related to the low bacterial titer present in the phyllosphere of weeds during the evaluation period. Other factors, such as bacterial composition in the phyllosphere, species, leaf type, phenological stage, and climatic conditions, may also have influenced the epiphytic colonization [35–37]. Leaf type is one of the factors that can influence the dynamics of epiphytic populations [38,39]. Leaves with trichomes tend to host larger bacterial populations than those with waxy cuticles [38]. Another factor to be considered during bacterial hosting is the availability of nutrient sources in the leaves, such as sugars, which can regulate the size of the epiphytic population [30].

Scanning electron microscopy (SEM) analyses enabled visualization of the epiphytic colonization of *Xvv* in several potential hosts. SEM showed extensive colonization of *Xvv* in the presence of exopolysaccharides (EPS) in maize cv. IPR 164, *A. sativa*, and *A. strigosa*. Studies have suggested that EPS have several functions, including bacterial protection against plant bacteriostatic substances and reduced direct contact with plant cells to minimize host defense responses [40]. Moreover, EPS promote water retention in plant tissue, which stimulates tissue rupture, allowing movement, invasion, or colonization by phytopathogens from distant sites [40–42]. In the genus *Xanthomonas*, xanthan is an EPS secreted by *Xvv* [43]. This EPS is involved in several functions, such as suppression of the plant's local defense and inhibition of callose deposition promoting pathogen dissemination [44–46]. In previous studies of *X. citri* subsp. *citri* (*Xcc*), xanthan was observed acting during the later stages of the infection process, probably increasing bacterial adherence to the host tissue and epiphytic survival, thus facilitating colonization in long-distance tissues [47].

Although we evaluated a broad range of genotypes in this study, 58 different plants, additional investigations are required to determine whether other mono- and dicotyledonous cultivated plants or those that occur in maize-growing areas can potentially host *Xvv*. Our findings revealed that several cultivated plants, cover crops, forage, and weeds can host *Xvv* pathogenically and/or epiphytically. For this reason, rational use of these plants within an integrated disease-management program is needed to minimize possible

BLS outbreaks in maize-producing areas. Practices using green manure and crop rotation should be conducted carefully, whenever possible, with non-host species for *Xvv* in areas with BLS incidence.

## 5. Conclusions

This is the first study conducted in Brazil to report alternative hosts for *Xanthomonas vasicola* pv. *vasculorum* (*Xvv*). Based on our findings, in addition to maize, several other symptomatic host plants of *Xvv* were found, including *Avena sativa*, *A. strigosa*, *Brachiariabrizantha* (Brizantha and cv. Marandu), *Digitaria horizontalis*, *D. insularis*, *Echinochloacolonum*, *Eleusine indica*, *Hordeum vulgare*, and *Sorghum arundinaceum*. Moreover, the phyllospheres of several species of cultivated plants, cover crops, forages, grasses, and weeds are suitable niches for the epiphytic survival of *Xvv*, which must be considered during management programs for bacterial leaf streak (BLS) control.

**Author Contributions:** Conceptualization and methodology, R.P.L.J., M.I.B.-P. and T.V.L.; formal analysis and investigation, T.V.L., R.R.R. and D.U.d.C.; scanning electron microscopy (SEM) analysis, T.V.L. and A.G.d.O.; resources and funding acquisition, R.P.L.J. and M.I.B.-P.; data curation and writing—original draft preparation, T.V.L. and D.U.d.C.; writing—review and editing, R.P.L.J. and M.I.B.-P. All authors have read and agreed to the published version of the manuscript.

**Funding:** This research was funded by Coordenação de Aperfeiçoamento de Pessoal de Nível Superior (Capes, Grant No. 88882.448321/2019-01), Instituto de Desenvolvimento Rural do Paraná, IAPAR/Emater (IDR-Paraná), Superintendência de Ciência, Tecnologia e Ensino Superior (SETI) do Paraná, Fundação Araucária (FA, Chamada 03/2021) and Universidade Estadual de Londrina (UEL).

**Data Availability Statement:** All data generated and analyzed during this study are presented in the published version of this article.

**Acknowledgments:** The authors would like to thank the staff and faculty of UEL and IDR-Paraná for their technical support. T.V.L. thanks the Capes for scholarship, IDR, SETI, FA and UEL for partial support for publication costs.

**Conflicts of Interest:** The authors declare no conflict of interest.

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
