# Peer review of "New Insights on Alternative Hosts of Xanthomonas vasicola pv. vasculorum, the Causal Agent of Bacterial Leaf Streak of Maize"

_agronomy, doi:10.3390/agronomy13041073_

Round 1

Reviewer 1 Report

The research paper entitled ‘New insights on alternative hosts of Xanthomonas vasicola pv. vasculorum, the causal agent of bacterial leaf streak in maize”  by Longhi et al. fits the scope of the Agronomy journal. The manuscript reports the survey of the infection of alternative hosts by X. vasicola pv. vasculorum. Although the authors somehow repeated a study conducted six years ago (Hartman et al., 2020), they expanded it by including new plant species that had not been tested before.

There are a few shortcomings to be addressed prior to publication.

First, the absence of statistical analysis of the obtained results. Pathogenicity data can be subjected to analysis of variance.  Is it possible to add information about disease incidence and disease severity? Have the differences in Xvv microbial counts (log CFU/g ) among the  various plants been statistically significant?

Lines 76-77: Table 1 does not contain 52 plant species.

In order to make Tables 1, 2 and 3  more readable, wouldn’t it be better to merge them and add a column for plant categories (e.g. crops, forage, grasses, weeds). Highlight IPR 164 and IPR 127 maize hybrids as patterns of susceptible hosts (positive controls). Table 3: “Finger grass” -> additional dot.

Line 85: more detail how the soil was prepared, whether it was sterilized?

Line 113: whether the number of bacteria in the suspensions was confirmed only by measurement of the optical density or by planting of serial dilutions of the cell suspensions?

Line 114: “… by spraying until complete runoff” -> Can you tell how many ml of bacterial suspension were sprayed approximately?

Line 120: “… were repeated a minimum of two times” -> Does it mean that some experiments have been carried out 3 or more times? If so, which one and why?

Line 123: Why is the publication [12] cited as referring to Koch's postulates, although it doesn't say anything about these postulates?

Line 130: add manufacturer of NA medium

Line 131: “Drigalski loop” -> “Drigalski spreader”, “Drigalski spatula”?

Line 146: CFUs (Log10·g −1 ) -> log CFU·g −1 ?

Line 146: “Five to ten colonies …” -> Any colonies or only suspected to be Xvv colonies?

Line 150: more detail about bacterial suspension (how many colonies, suspension in water or buffer, how many ml)?

I suggest to boost the quality of Figures 2 and 3 (e.g. now unnecessarily large size, different font style of subfigure captions (A, B, C etc.)

Line 495: ”… xanthan is secreted by EPS” -> xanthan is an EPS secreted by Xvv

Author Response

Dear editor we really appreciated all the suggestions and comments stated by the Reviewers. All the modifications stated by them were of utmost importance to improve the quality of our manuscript. Please find in the below the questions and comments made by the Reviewers, all of them were addressed in the text.

Reviewer 2 Report

In this study, alternate hosts of bacterium Xanthomonas vasicola pv. vasculorum (Xvv) causing Bacterial leaf streak (BLS) of maize (Zea mays) was studied. Study of parasitism and epiphytic colonization of the bacterium in asymptomatic plants was undertaken. Some good results have come out of the study. As propagule survival is an important source of inoculum for next season crops the knowledge of such host is important to plan a strategy for management. The publication contains good information but needs to address the following points

Comments

The introduction could be improved. Importance of alternate host for Xanthomo-39 nas  vasicola pv. vasculorum (Xvv) and that too in dicots should be emphasized more with some reference. Importance of perennial forage and weed plant as alternate host shouls be emphsised

Line

77: clarify certain “species”

106-107: Specify the region of the bacterial genome that was used to design the primer

113: That OD was measured by a spectrophotometer of the make. at particular wavelength should be mentioned. Only the model of spectrophotometer is mentioned. This makes the sentence incomplete and difficult to understand.

134: Explain or rephrase “PCR in at least one of 134 the experiments”

136 -138: Explain or rephrase “and at 15, 21, and 30 DAI in the second experiment”.

141:  Mention the instrument used for stirring

142: Explain the “sample”

143: Explain “deposited in Petri dishes”

161: Not required “(50X TAE: Tris, glacial acetic acid, and EDTA)”

Table 1 :  In the table symbols are -S1,           EC2  but in caption it is 1S ,2EC  Are they same?  Then it should be written in same manner.  

Line

208: “ Xvv survived epiphytically for up to 30 days” but the table 4 indicates the maximum recovery period with significant log values. Does the maximum recovery period figure given in the table indicates that highest (quantitatively) CFU is recovered by that day and then declines? Or it is recovered till that day and none is found after that day?

214: “The results indicated variation” can be changed to “Results also indicated variation”

409-410: Rephrase it to make the meaning clear“evaluated under field conditions naturally  infested with Xvv”

453: “are perennial species; therefore, they” semicolon and comma should be removed

Discussion is written well but major findings should be highlighted.

Line

455-464 could be written more conclusively using more references indicating he importance of weeds and forage plants as alternate host

476-486- not concluded properly

Author Response

Response letter

Dear editor we really appreciated all the suggestions and comments stated by the Reviewers. All the modifications stated by them were of utmost importance to improve the quality of our manuscript. Please find in the below the questions and comments made by the Reviewers, all of them were addressed in the text.

Reviewer 3 Report

This is an interesting and well-written, well-presented and well-executed study on the host range of a Brazilian strain of Xvv. The results are important. The major limitation is that the results are for just a single strain of Xvv and it would have been interesting to systematically investigate whether there are variations in host range among strains of Xvv. IT would also have been interesting to test eucalyptus. However, that would have required a large amount of extra work. 

I have just a few small suggestions for possible improvement and for additional information that might be interesting to the authors.

(1) The title is "New insights ...". It would be much better to clearly and concisely state the most important new insight in the title, like a newspaper headline.

(2) Lines 50-55. The authors should be aware that there is some confusion and ambiguity about the causal agents of gumming disease.  What was previously called X. campestris pv. vasculorum has now been split into two very distinct species of Xanthomonas. This is reviewed in https://doi.org/10.1094/phyto-03-19-0098-le. 

(3) Lines 395-399. The authors should be aware that there are claims that Xvv can cause disease on Eucalyptus. See: https://doi.org/10.1111/ppa.12298.

(4) Lines 424-425. When discussing previous work on host range of Xvv, again it might be useful to read https://doi.org/10.1094/phyto-03-19-0098-le to find citations of previous work that define the host range of Xvv. Also, one of our papers that the authors missed is https://doi.org/10.5897/AJPS2015.1327. I am not asking the authors to cite our work, but they should at least be aware of it.

(5) Lines 424-435. It is not a requirement for the current paper, but I would strongly request that the authors sequence the genome of their RL1 strain and deposit this in the public databases and perhaps publish a genome announcement. There is value in having a genome sequence for phenotypically well-characterised strains. 

(6) Lines 424-434. Again, I do not request citation, but I would like to draw the authors' attention to our previous work on comparative genomics of Xvv, which may provide some useful context and background: https://doi.org/10.3390/pathogens3010211.

Author Response

(The authors gave the same response as above.)
